# Targets for Renal Carcinoma Growth Control Identified by Screening FOXD1 Cell Proliferation Pathways

**DOI:** 10.3390/cancers14163958

**Published:** 2022-08-16

**Authors:** Kyle H. Bond, Sunder Sims-Lucas, Leif Oxburgh

**Affiliations:** 1Rogosin Institute, Room 2-43, 310 East 67th St., New York, NY 10065, USA; 2Children’s Hospital of Pittsburgh, Rangos Research Building, 4401 Penn Ave, Pittsburgh, PA 15224, USA

**Keywords:** kidney cancer, FOXD1, FOXM1, PME1, TMEM167A, FDI-6, AMZ30, silibinin, cell cycle, bioengineering

## Abstract

**Simple Summary:**

FOXD1 regulates the proliferation of clear cell renal cell carcinoma (ccRCC) cells, and ccRCC cells in which *FOXD1* has been inactivated do not form tumors efficiently in an animal model. Reproducing growth inhibition in tumor cells by inhibiting FOXD1 pathways presents a possible therapeutic approach for ccRCC and other cancers. We have established an analysis strategy to identify *FOXD1*-regulated target pathways that may be therapeutically tractable, and compounds that modulate these pathways were selected for testing. Targets in three pathways were identified: FOXM1, PME1, and TMEM167A, which were inhibited by compounds FDI-6, AMZ-30, and silibinin, respectively. The effects of these compounds on the growth of tumor cells from patients cultured in a novel 3D tumor-replica culture environment revealed that FDI-6 and silibinin had strong growth inhibitory effects. This investigation informs new therapeutic targets to control ccRCC tumor growth, and provides a strategy to compare the responsiveness of individual patient tumor replicas to growth-inhibitory compounds.

**Abstract:**

Clinical association studies suggest that FOXD1 is a determinant of patient outcome in clear cell renal cell carcinoma (ccRCC), and laboratory investigations have defined a role for this transcription factor in controlling the growth of tumors through regulation of the G2/M cell cycle transition. We hypothesized that the identification of pathways downstream of FOXD1 may define candidates for pharmacological modulation to suppress the G2/M transition in ccRCC. We developed an analysis pipeline that utilizes RNA sequencing, transcription factor binding site analysis, and phenotype validation to identify candidate effectors downstream from FOXD1. Compounds that modulate candidate pathways were tested for their ability to cause growth delay at G2/M. Three targets were identified: FOXM1, PME1, and TMEM167A, which were targeted by compounds FDI-6, AMZ-30, and silibinin, respectively. A 3D ccRCC tumor replica model was used to investigate the effects of these compounds on the growth of primary cells from five patients. While silibinin reduced 3D growth in a subset of tumor replicas, FDI-6 reduced growth in all. This study identifies tractable pathways to target G2/M transition and inhibit ccRCC growth, demonstrates the applicability of these strategies across patient tumor replicas, and provides a platform for individualized patient testing of compounds that inhibit tumor growth.

## 1. Introduction

Clinical association studies and laboratory investigations have identified the FOXD1 transcription factor as a driver of tumor development. Understanding how this may be targeted has implications for a variety of cancers [1,2,3,4]. Due to the central role *Foxd1* plays in regulating the embryonic development of the kidney [5,6], we are interested in understanding its role in clear cell renal cell carcinoma (ccRCC), which is the most common form of kidney cancer. Renal cell carcinoma represents 3% of all tumors, with 75% of those being ccRCC. Primary care for non-metastatic disease includes resection, while metastatic disease involves combinatorial adjuvant therapy using tyrosine kinase inhibitors (TKIs) and immune checkpoint inhibitors [7].

In previous work, we showed that *FOXD1* expression correlates with poor patient outcomes in ccRCC, that its inactivation in ccRCC cells delays progression through the G2/M phase of the cell cycle, and that *FOXD1* loss prevents tumor growth in a xenograft model [8]. Phosphorylation of CDC2 and histone H3 is perturbed by the loss of *FOXD1*, suggesting that FOXD1 expression is an adaptation in tumor cells that promotes their proliferation by enabling their progression through the G2/M checkpoint. *FOXD1* null tumor cells display aberrant cytokinesis and limited proliferation; replicating this phenotype in cancer cells presents an interesting therapeutic approach for ccRCC, and possibly other cancers. However, few strategies have been reported to directly target transcription factors in tumor cells. In the case of FOXD1, very little is known regarding its molecular functions, such as regulation by upstream kinases or interactions with other transcription factors, making it a particularly difficult target. A more tractable strategy is to modulate pathways regulated by FOXD1, and in this study we used a transcriptomic analysis to identify specific pathways downstream from FOXD1 that can be pharmacologically modulated to suppress the G2/M transition in ccRCC tumor cells.

An analysis pipeline was developed that utilizes RNA sequencing, transcription factor binding site analysis, and phenotype validation to identify candidate effectors downstream from FOXD1. We identified candidates with known functions related to the *FOXD1* null phenotype, and compounds that modulate their function were evaluated for their potential to replicate the cell cycle phenotype caused by the loss of *FOXD1*. A panel of patient-derived 3D tumor replicas was developed using a method recently described by our group to validate the effects of candidate compounds in clinically relevant models [9]. Our study uncovers FOXM1, PME1, and TMEM167A as potential targets for pharmaceutical intervention to slow ccRCC tumor cell growth and provides a strategy for patient-specific compound testing.

## 2. Materials and Methods

### 2.1. Cell Lines

The 786-O cell line was obtained from ATCC (ATCC CRL-1932). The 786-O^FOXD1null^ line was generated in our laboratory, as previously reported [8]. Cells were maintained in RPMI-1640 containing 10% FBS, 1% GlutaMax (Thermo Fisher, Waltham, MA, USA), and 1% Penicillin-Streptomycin, and were grown to 70–80% confluency on tissue culture-treated plates before experiments. Cells were detached from the plates using TrypLE Express (Thermo Fisher).

### 2.2. RNA Sequencing and Data Acquisition

RNA from 70% confluent plates of 786-O and 786-O^FOXD1null^ cells was extracted using a Qiagen RNeasy Microkit (Qiagen GmbH, Hilden, Germany). RNA purity was determined by measuring the 260/280 ratio in a Thermo Fisher Nanodrop. RNA quality was determined by running an RNA gel and analyzing the ratio between 28 S and 18 S. Samples with a 28 S/18 S ratio greater than 2.0 were submitted for RNA sequencing by Genewiz (Chelmsford, MA, USA). Pre-made total RNA libraries were prepared and sequenced using a HiSeq 2000 in 2 × 150 bp sequencing mode. Quality was assessed per base sequence, and mean quality was calculated. Quality scores greater than 30 were deemed acceptable for further downstream analysis. Primer and adaptor sequences were trimmed using Trimmomatic v.0.36, and only high-quality reads with ≥30 bp over a 4 bp sliding window based on the phred algorithm were accepted. Reads were aligned to the human genome build hg38, using STAR aligner v.2.5.2b with default settings for mammalian genomes. Unique gene hit counts were calculated using the featureCounts function from the Subread RNA-seq package v.1.5.2. This did not include intron or intergenic regions and used Gene ID as the identifier.

### 2.3. Gene Set Enrichment Analysis

Differentially expressed genes from RNA sequencing were ranked for fold change and analyzed using a gene set enrichment analysis (GSEA) [10]. Pathway interactions were mapped using the Reactome database [11]. Pathways and interactors were mapped using Cytoscape [12].

### 2.4. RNA-Seq Data Analysis

Uniquely mapped reads were analyzed for differential gene expression using DESeq2, comparing 786-O and 786-O^FOXD1null^. Fold change and *p*-values were calculated using the Wald test. Only adjusted *p*-values less than or equal to 0.05 were considered “differentially expressed”. Genes without *p*-values were deemed “uninterpretable” and discarded from further analysis. All other genes were labeled “equally expressed”. Differentially expressed genes were divided into upregulated (foldchange >1) or downregulated (foldchange <1), and the top 100 most up- or downregulated genes were set aside for scoring. In parallel, the entire list of differentially expressed genes was analyzed for transcription factor binding sites using CiiiDER [13]. Briefly, transcription factor matrices from the JASPAR2020 database were mapped to the input gene list with the gene look-up manager (GLM) from the human genome atlas build hg38, using a deficit cutoff of 0.15 (15%). Genes with FOXD1 motifs were further analyzed for “clustered FOXD1” or FOXD1 binding motifs within 100 bp from each other. These “clustered FOXD1” targets were separated for further analysis as candidate FOXD1-regulated genes.

### 2.5. Conditioned Media Growth Rate Analysis

The 786-O and 786-O^FOXD1null^ cells were seeded into 15 cm dishes and allowed to grow to confluency. At confluency, the culture media was replaced and cultured for an additional 48 h. Conditioned media was collected and centrifuged at 14,000× *g* for 20 min at 4 °C to pellet any cell debris. Supernatant was collected and stored at 4 °C. For the growth assay, 5000 cells per well of 786-O or 786-O^FOXD1null^ cells were seeded into 24-well tissue culture plates in either conditioned media from 786-O or 786-O^FOXD1null^. Every day after seeding, cells were collected from triplicate wells and counted. Media in remaining wells was changed daily. This was undertaken until day 7 and cell numbers were graphed using Excel.

### 2.6. Native ECM Growth Rate Analysis

Plates covered in native ECM from 786-O and 786-O^FOXD1null^ were derived following the protocol from Harris et al. [14]. In brief, 786-O or 786-O^FOXD1null^ were seeded at confluency into 24-well plates, then cultured for 5 days. Plates were washed twice with DPBS, and 3 times with wash buffer 1 (100 mM Na_2_PO_4_, 2 mM MgCl_2_, 2 mM EGTA, pH 9.6). Cells were lysed twice with a lysis buffer (8 mM Na_2_PO4, 1% NP-40, pH 9.6) for a total of 1 h and 30 min at 37 °C. Plates were then washed 3 times with wash buffer 2 (10 mM Na_2_PO_4_, 300 mM KCl, pH 7.5) and finally washed 4 times with sterile DI water. Plates were stored filled with DPBS at 4 °C until the assay. At the start of the assay, 70% confluent plates of 786-O and 786-O^FOXD1null^ were detached using TrypLE express, counted, and seeded into plates containing native ECM from 786-O or 786-O^FOXD1null^ at 5 × 10^4^ cells per well. Every 24 h after seeding, triplicate wells were imaged, detached, and counted. Media was changed daily in all other wells. This was undertaken until day 7 and cell numbers were graphed using Excel.

### 2.7. Target Gene Selection

Genes selected from the RNA-seq analysis (top 100 up- and downregulated genes, genes with unclustered FOXD1 motifs from 15% cutoff analysis) were cross-referenced with the following databases: Mitotic G2-G2/M phases (Reactome R-HAS-453274) [15], Human kinome (Kinbase) [16], Matrisome [17], and Cell Adhesion (GO: 0007155) [18]. Scores were weighted as follows: Mitotic G2-G2/M phases = +2, Human kinome = +1, Matrisome = +0.5, Cell Adhesion = +0.5. Genes that scored greater than or equal to a final score of 2 were assessed further for targetability. All candidates were divided into requiring either inhibition (log fold change >0) or activation (log fold change <0). Targets requiring inhibition were first searched online as follows: “Protein Name” + (“Inhibitor” OR “Antagonist”). If a specific inhibitor exists, the search ended, and the compound was added to the panel. For targets requiring activation, the Uniprot protein description was reviewed to identify if the protein was secreted or not. If yes, and a recombinant exists, it was added to the candidate panel. If not, an online search was conducted as follows: “Protein Name” + (“Activator” OR “Agonist”). If a specific activator or agonist exists, then it was added to the candidate panel. The final panel was further refined to remove secreted growth factors, extracellular matrix (ECM) components, and targets with known or contradictory effects based on the literature. All other targets from this analysis were not explored further in this study.

### 2.8. Compound Toxicity Screening

786-O cells were seeded into 96-well plates at 2 × 10^4^ cells per well and cultured for 2 h at 37 °C. The 5- to 6-point dose-responses were determined based on the following: 1. IC50 (1×, 2×, 3×, 4×, 5×, and 50–100×), 2. ED50 (1×, 2×, 3×, 4×, 5×, and 50–100×), or 3. from reported working concentrations. Exact titrations are summarized in Appendix A. Selected target compounds were prepared according to manufacturer specifications, diluted to selected dose ranges in the culture media, and added to wells in triplicate. Vehicle-treated controls were included in each plate. After 3 days, wells were stained with Hoechst 33342 and Ethidium Homodimer-1 for 30 min at room temperature. Wells were imaged using a Thermo Fisher EVOS microscope. Images were quantified for number of cells per well (# Hoechst 33342 stained nuclei) and number of dead cells (# Ethidium Homodimer-1-stained nuclei). Percent growth inhibition was calculated based on the average number of cells per well compared to untreated wells. Percent viability was calculated based on the average number of Ethidium Homodimer-1 negative cells to the average total number of cells. Pairwise comparisons between treatments and untreated wells were conducted using a Student’s *t*-test, with *p*-values less than 0.05 designated as significant. The optimal dose per compound was chosen based on the greatest amount of inhibition without toxicity.

### 2.9. Synchronized Cell Cycle Analysis

The 786-O and 786-O^FOXD1null^ cells were seeded into culture treated dishes at 10.5 × 10^3^ cells per cm^2^ and cultured overnight at 37 °C. Cells were then treated with 2.5 mM thymidine in culture media for 16 h, followed by a media change and retreatment with 2.5 mM thymidine for an additional 9–12 h. One plate was left untreated as a control. Plates were then washed with DPBS 2 times before adding the culture media. One synchronized plate was collected and fixed at this point as a control. For treatments, media containing the different compounds selected from our dose-response analysis was added instead. Cells were cultured for 11 h at 37 °C and then lifted off the plates with TrypLE, centrifuged, and resuspended in 100 µL of the culture media. Following this, 900 µL of freezing 100% methanol was added drop-wise and cells were transferred to a −20 °C freezer and fixed for at least 2 h. Afterwards, cells were washed with MACS flow buffer (Miltenyi Biotec, Bergisch Gladbach, Germany) 3 times, then resuspended in 1 mL MACS flow buffer containing 50 µg/mL DAPI and incubated at 4 °C on a nutator overnight. Cells were washed 3 times with MACS flow buffer, resuspended in 400 µL of MACS flow buffer, and analyzed using Miltenyi MACSquant VYB flow cytometer. The G1 phase was determined by analyzing synchronized controls, in which all cells are in G1. The G2 phase was determined by doubling the value of suspected G1, while the S phase was all values in between.

### 2.10. 3D Cultures

Three-dimensional fibrin cultures of 786-O, 786-O^FOXD1null^, and primary RCC cell lines were created following our previously reported protocol [9]. In summary, 3D cultures were generated by combining fibrinogen (MilliporeSigma, Burlington, MA, USA cat#F8630) at a 1:1 ratio with cells suspended in a culture media, with or without the addition of ECM proteins. Thrombin (MilliporeSigma cat#10602370001) was added at a concentration of 0.002 U/µL at a volume equal to 10% of the total cell-fibrinogen mix, then immediately spotted onto glass coverslips at 5 µL per replicate. Three-dimensional cultures were immediately placed into a 37 °C cell culture incubator for 30 min to facilitate gelation. Pre-warmed cell culture media, with or without treatments, was then carefully added to wells.

### 2.11. Live/Dead Analysis

After 3 days of culture, the media was replaced with a live/dead staining solution containing 2 μM Calcein-AM and 4 μM Ethidium homodimer-1 (Etd1) in the culture media (Thermo Fisher cat#L3224) and incubated for 30 min in a culture incubator. In the final 8 min, Hoechst 33342 (Thermo Fisher cat# H3570) was added at a concentration of 5 µg/mL to each sample. Images were collected on an EVOS imaging system. Images were analyzed for Calcein-AM positivity to identify the number of cellular structures. Each structure was designated as either live (Calcein+) or dead (Ethd2+). Higher magnifications were taken to analyze the nuclear structure stained with Hoechst 33342. Comparisons between culture conditions were done via z-test.

### 2.12. 3D Culture Structure Analysis

Structures from images were analyzed using FIJI [19]. First, all channels were binarized using the threshold tool. Calcein images, which include the whole body of the cells, were used as regions of interest (ROIs). Descriptive statistics, including number of structures and area of each structure, were calculated using the measure tool. Comparisons between conditions were analyzed using a Student’s *t*-test and z-test.

### 2.13. Nuclear Size Analysis

Nuclei from images were analyzed using FIJI. In brief, Hoechst 33342 stained nuclei were binarized using the threshold tool, the background was subtracted with the despeckle tool, and they were separated using the watershed tool. The pixel area of each nucleus was measured using the measure tool. Descriptive statistics were performed and comparisons between conditions was performed using a Student’s *t*-test. Over 500 nuclei were analyzed per condition.

### 2.14. Tissue Processing

Kidney tumor tissue was obtained from surgical nephrectomies performed at the University of Pittsburgh Medical Center. Cancer diagnosis was performed by certified pathologists. Voluntary informed consent was obtained from patients for use of their tumors and diagnostic results in this research. All tissue specimens and associated medical information received were de-identified using an honest broker system. Samples were processed as previously described [9]. Fresh tumor tissue samples were diced and weighed. A total of 100 mg of tissue was digested in 20× volume DMEM containing 250 U/mL Collagenase IV (Thermo Fisher) + 0.02% Trypsin-EDTA at 37 °C in a shaking incubator set to 200 RPM for up to 2 h or until media became cloudy. The digest was stopped by adding an equal volume of ice cold DMEM containing 10% FBS. Any undigested material was filtered with 100 µM filter and placed in a vessel containing 5× volume TrypLE Express (Thermo Fisher) and returned to the 37 °C shaking incubator set to 220 RPM for an additional 15 min. Samples were pooled if necessary and centrifuged at 500× *g* for 7 min 30 s at 4 °C. Cells were washed with ice cold DMEM, counted, and their viability was determined using Trypan Blue exclusion. This was utilized for downstream assays or frozen in Nutrifreeze (Sartorius, Göttingen, Germany) freezing media. Patient samples used in this study are summarized in Table 1.

### 2.15. Primary Cell Culture

De-identified kidney tumors obtained through voluntary informed consent (as described in 2.14) were processed following our previously reported protocol [9]. Digested tumor cells were washed and resuspended at 1 × 10^6^ cells/5 mL High Glucose DMEM containing 10% FBS, 1% GlutaMax (Thermo Fisher), 1% Penicillin-Streptomycin, 1% Non-Essential Amino Acids (Thermo Fisher), and 1% Sodium Pyruvate, and plated onto 5 cm tissue culture treated plates. Plates were incubated at 37 °C for 24 h before media change and imaging. Cells were monitored with media change every 72 h. After reaching 70–80% confluency, cells were expanded into a 10 cm tissue culture treated dish. Upon reaching 70–80% confluency, passaged cells were either frozen in freeze media or expanded for downstream applications. Three-dimensional cultures were established as previously reported [9].

### 2.16. Western Blot Analysis

Protein was collected from cell lines using a 1× Laemmli Buffer. Western Blotting was performed following manufacturer guidelines using the BioRad Western Blotting system (BioRad, Hercules, CA, USA). Blots were stained for beta-tubulin (Thermo Fisher MA5-16308), VHL (Cell Signaling Technologies, Danvers, MA, USA #68547), and HIF1α (R&D Systems, Minneapolis, MN, USA, AF1936). Molecular weight and densitometry analyses were performed using BioRad Imagelab software and normalized to beta-tubulin density.

## 3. Results

### 3.1. RNA-Seq Analysis Identifies Novel Targets Downstream of FOXD1

We have previously demonstrated that loss of FOXD1 in 786-O ccRCC cells causes impairment of the cell cycle, severely limiting their proliferation [8]. Knockout of *FOXD1* perturbs phosphorylation of the key cell cycle proteins CDC2 and histone H3 and dysregulates the G2/M transition. Comparison of 786-O and 786-O^FOXD1null^ transcriptomes has the potential to reveal transcriptional networks essential for G2/M control that may yield therapeutic targets; these could be either direct transcriptional targets of FOXD1, or genes in pathways downstream from direct targets that are mis-regulated by the loss of FOXD1.

Comparison of 786-O with 786-O^FOXD1null^ by RNA-seq revealed 5828 differentially expressed genes using a false discovery rate (FDR) and an adjusted *p*-value cutoff of 0.05. To prioritize a subset of these genes for downstream analysis, we used the selection pipeline shown in Figure 1. For our most stringently selected subset of potential direct FOXD1 targets, we identified genes with a FOXD1 binding motif (15% tolerance for mismatch of bases outside the core consensus sequence of the extended TFBS matrix) located within 1.5 kilobases upstream or 0.5 kilobases downstream of the locus [20]. Predicted FOXD1 binding motifs in differentially expressed genes were mapped using the CiiiDER tool for predicting and analyzing transcription factor binding sites, resulting in a list of 4298 genes [13]. Clustering of transcription factor motifs associates with transcriptional control [21], and we identified a subset of 23 genes with two or more predicted FOXD1 motifs within 100 bases. This high stringency subset of differentially expressed genes were selected for downstream analysis (Appendix A).

In a second selection strategy, differentially expressed genes were manually cross-referenced to four databases representing cellular phenotypes that are perturbed following *FOXD1* inactivation, with the rationale that mis-regulation of any of these pathways may impair the G2/M transition. Databases included Mitotic G2-G2/M phases (Reactome R-HAS-453274) [15], Human kinome (Kinbase) [16], Matrisome [17], and Cell Adhesion (GO: 0007155) [18]. The top 100 upregulated and top 100 downregulated genes, based on log fold change and the remaining 4275 differentially expressed genes with predicted FOXD1 binding motifs, were screened in this manner. To weight our gene selection in favor of G2/M, we used the scoring system shown in Figure 1 to rank candidates. The 92 top-ranked genes were selected, and together with the 23 candidates with clustered FOXD1 binding sites, this generated a list of 110 candidates for downstream analysis (Appendix A).

### 3.2. Exclusion of Secreted Factors

Candidates included genes in secreted signaling pathways, suggesting that autocrine signaling may be modified in *FOXD1* null cells. To test this hypothesis, we collected and purified conditioned media from both 786-O and 786-O^FOXD1null^ and performed a growth rate analysis of cells grown in their own conditioned medium or in the medium of the other line (Figure 2A). Conditioned medium from either line did not reduce growth, indicating that changes in secreted factor signaling are not the cause of growth inhibition caused by the loss of FOXD1, and secreted factors were excluded from further analysis.

### 3.3. Exclusion of ECM

Candidates also included genes involved in formation and deposition of ECM. Appropriate attachment substrates can strongly influence cell division [22], and to test if aberrant ECM deposited by *FOXD1* null cells affects cell growth, we coated plates with native ECM from 786-O or 786-O^FOXD1null^ by decellularizing confluent plates of 786-O ECM or 786-O^FOXD1null^ as previously described [14]. We found that 786-O cells were unaffected by growth on 786-O^FOXD1null^ ECM, suggesting that changes in ECM composition in the 786-O^FOXD1null^ do not cause growth impairment. Interestingly, 786-O^FOXD1null^ were growth inhibited when cultured on 786-O ECM (Figure 2B). This series of experiments shows that the ECM deposited by 786-O^FOXD1null^ cells is itself not growth inhibitory, and based on this observation, we excluded ECM from further analysis.

### 3.4. Selection of Activators and Inhibitors

For the remaining 73 candidate genes, we manually searched for compounds that could be used to modulate their function in vitro. Since directionality of gene expression change was not used as a criterion for candidate gene selection, it was necessary to search both for inhibitors and activators, including recombinant proteins. Twelve candidate compounds that inhibit the products of genes downregulated in 786-O^FOXD1null^ and six candidate compounds that activate products of genes upregulated in 786-O^FOXD1null^ were selected for analysis for their potential to phenocopy the G2/M delay seen with the loss of FOXD1 (Table 2).

### 3.5. Inhibition of Cell Growth with Minimal Toxicity by Compounds Targeting Candidate Genes

To assess if any of these targets can mimic the phenotype seen in 786-O^FOXD1null^, we screened all candidate compounds in a dose-response study. Doses were chosen based on IC50, EC50, or literature-reported concentrations summarized in Appendix A. Cells were treated for three days to assess growth inhibition and toxicity, which we defined as a significant increase in cell death compared with the vehicle-treated control. At the end-point, wells were stained for the nuclear marker Hoechst 33342 and dead cell marker Ethidium homodimer-1. Cell number per well was counted and percent cell death was calculated. As our goal was to replicate the *FOXD1* null G2/M phenotype, our target for each compound was the concentration at which cell number was reduced without overt cell death, indicating an effect on proliferation rather than cytotoxicity. To control for the possibility that the viable cell count may be lowered by cell death prior to harvest, we quantified death in all cells in the culture including those floating in the medium. Of the tested compounds, 10 showed significant reduction in cell growth compared to untreated controls, without significant toxicity (93 ± 5% viable) (Figure 3).

### 3.6. Selection of Candidate Compounds That Induce Cell Cycle Arrest Resembling the Effect of FOXD1 Inactivation

To determine if growth reduction is due to growth arrest at G2/M, we synchronized cells using the double thymidine block method. We previously showed that after synchronization, it took 786-O cells 12 h to complete the cell cycle, while 786-O^FOXD1null^ required 14 h [8]. After synchronizing cells, we allowed them to proceed through the cell cycle and treated them with the maximal tolerated dose of each compound that was shown to inhibit growth without significant effect on viability. Cells were cultured for 10 h, at which point we measured the ratio between cells at G1 and G2/M to assess if any compound recapitulates 786-O^FOXD1null^ growth delay (Figure 4). The 786-O^FOXD1null^ cells were included in the analysis for direct comparison. Three compounds showed an increase in distribution of cells at G2/M when compared to 786-O 11h after lifting the thymidine block: 3 µM AMZ30, 135 µM FDI-6, and 247.2 µM silibinin. Others showed effects on the S-phase transition (0.8 µg/mL recombinant ANGPTL4, 24 µM APE inhibitor III, 2 µM BITC, 0.2 µg/mL recombinant SHH), but were excluded from further analysis as our focus was on reproducing the effect of *FOXD1* loss on G2/M.

### 3.7. FOXD1 Inactivation Affects Three-Dimensional Growth

Three-dimensional (3D) culture provides a cellular environment with higher physiological fidelity than monolayer growth, particularly with regard to growth and proliferation characteristics. To extend our analysis of the growth-limiting effects of compounds selected in our screen, we established 3D growth models using a cell culture system developed in our laboratory to mimic the specific ECM environment of ccRCC in 3D [9]. In brief, cells are scaffolded in a fibrin hydrogel containing a combination of nine ECM components characteristic of ccRCC including fibronectin, collagen 6, and tenascin C, where their invasive versus colony-forming growth can be followed by light microscopy. The 786-O cells displayed an invasive phenotype in the 3D culture system, whereas the 786-O^FOXD1null^ cells grew in round, expansive clusters, and did not display an invasive phenotype (Figure 5). We have previously shown that this 3D culture system can be used to generate replicas of ccRCC tumors from patients [2], providing an opportunity to test the potential of candidate compounds to inhibit growth of patient tumor replicas in addition to testing effects on characteristic 786-O 3D growth.

### 3.8. Treatment-Induced Growth Delay Affects 3D Growth and Nuclear Structure of 786-O Cells

Morphological differences between 3D-cultured 786-O and 786-O^FOXD1null^ included both changes in cell colony shape and changes in the shapes of individual nuclei within colonies. We used these two independent parameters to measure the effects of our candidate compounds on the 3D growth of 786-O cells. Three-dimensional cultures of 786-O cells were treated with 3 µM AMZ30, 135 µM FDI-6, and 247.2 µM silibinin, and were compared to 786-O^FOXD1null^. After three days of culture, cells were stained with the nuclear dye Hoechst 33342, the live-cell dye Calcein-AM, and the dead-cell marker Ethidium homodimer-1. Compared with control 786-O, compound-treated 786-O showed growth perturbations (Figure 6A–E). Cultures treated with AMZ30 and silibinin showed larger cell colony sizes (similar to 786-O^FOXD1null^), while FDI-6 had the opposite effect, promoting the growth of smaller and rounder cell colonies compared to the untreated cultures (Figure 6F). By quantifying the number of colonies in each 3D culture, we can see a significant increase in number following treatment with FDI-6 and silibinin, similar to 786-O^FOXD1null^ (Figure 6G).

Loss of *FOXD1* leads to an increase in cells at G2 with large nuclei, as well as an increase in cells with mitotic defects and nuclear fragmentation [9]. Considering that our compounds were selected to mimic the 786-O^FOXD1null^ phenotype, we hypothesized that the analysis of nuclear morphology of treated cultures would be informative (Figure 7A–F). Compared with 786-O, 786-O^FOXD1null^ cultures contain cells with larger nuclei (2.6-fold increase), with some evidence of nuclear defects (Figure 7A,B,F). Similarly, increased nuclear size was found following treatment with silibinin (1.36-fold increase) (Figure 7E). In contrast, treatment with AMZ30 and FDI-6 caused a reduction in nuclear size in the 786-O cells (1.4-fold and 2-fold decrease) (Figure 7C,D,F). In both treatments, nuclear mitotic defects were apparent in subsets of cells (115/186 nuclei: 62% fragmented) and with FDI-6, a large proportion of cells were anuclear, with darkened, protruding cell bodies (191/377 cells: 51%) (Figure 7D).

In summary, 3D growth and nuclear morphology of 786-O were strongly affected by inhibiting PME-1 (AMZ30), FOXM1 (FDI-6), and TMEM16A (silibinin), replicating aspects of the *FOXD1* loss of function phenotype in these cells. A summary of the findings is presented in Table 3. No one candidate entirely phenocopied 786-O^FOXD1null^; silibinin replicated most aspects of the phenotype, while FDI-6 had a distinct but severe effect on nuclear morphology. Our transcriptome analysis suggests that loss of *FOXD1* perturbs numerous pathways (Appendix A), and the finding that candidate compounds targeting individual pathways downstream from FOXD1 replicate discrete aspects of the phenotype is anticipated.

### 3.9. Effects of Blocking FOXD1 G2/M Targets on 3D Patient Tumor Replica Growth

To understand if the compounds inhibit the growth of primary tumor replicas from patients, we isolated seven cell lines from patient tumors following a published protocol [9]. Previous reports indicate that many cell lines grown out from *VHL*-mutant patient tumor samples do not harbor *VHL* mutations [49], which is contradictory to genetic studies showing that *VHL* is a truncal mutation in the majority of ccRCCs [50], and suggests that outgrowth of bystander untransformed cells may be a common phenomenon. Cell lines were grown to at least passage three, and protein was collected to assess VHL expression status. The 786-O cell has a mutated VHL and was used as a comparator. Of seven isolated lines, three lacked VHL (S109, S114, S438) and showed expression of HIF1α, indicating a pseudo-hypoxic state (Figure 8A). Conversely, two cell lines displayed strong VHL expression and no detectable HIF1α (S322, S453). Two cell lines showed intermediate VHL and HIF1α expression (S199, S277). From this analysis, five of seven cell lines display some degree of pseudohypoxia, and were categorized as tumor cells.

To determine whether these patient-derived cell lines could provide a testing platform for our candidate compounds, we cultured them in 3D and treated them with candidate compounds at the effective and non-toxic concentrations determined using 786-O cells (Figure 3). After three days of treatment, cultures were stained with Hoechst 33342, Calcein-AM, and Ethidium homodimer-1, and the number of viable cell colonies was compared between treatment conditions (Figure 8B,C and Appendix A). Three lines showed changes in the number of colonies when treated with AMZ30 (S114 1.2-fold decrease = 1.2; S199 1.2-fold decrease; S438 1.3-fold increase); interestingly only S114 showed a reduction in the number of overall colonies, while S199 and S438 both showed an increase. Four lines showed a strong response to FDI-6 (S109 1.6-fold decrease; S114 1.8-fold decrease; S199 1.3-fold increase; S438 1.3-fold decrease), all showing a reduction in the number of colonies, with the exception of S199. Morphological changes could be observed following FDI-6 treatment, with thinning of cells and pronounced cellular projections (Figure 8B and Appendix A). Additionally, we observed a loss of Hoechst 33342 staining in cells showing such morphological characteristics. Silibinin treatment reduced colony numbers in two lines (S109 1.9-fold decrease; S438 1.4-fold decrease). In summary, the concentrations of all three compounds derived by titration on 786-O cells did have significant effects on 3D cell growth in subsets of patient tumor cells.

Although effects on colony growth were seen with compound concentrations determined using 786-O cells, each patient line is unique and may have differing dose requirements. To assess if this was the case, we repeated the 3D-treatment assay with patient lines using a three-point range for each compound. Only FDI-6 showed a strong dose-dependent response with a number of cell colonies. However, this increase is also associated with an increase in nuclear defects (Appendix A).

Nuclear morphology changes were observed in 786-O cells treated with compounds, so we analyzed the nuclear morphologies of primary lines. Lines showed distinct nuclear size distributions (examples shown in Figure 8D), and thus analyses were controlled by comparison with vehicle-treated samples for each individual line. The 786-O^FOXD1null^ cell was controlled by comparison with the 786-O cell. Treatment with AMZ30 showed a modestly increased nuclear size in the S322 line (1.4-fold) at the dose predicted for 786-O, but not at any higher dose (Figure 8E,F). All lines tested showed dose-dependent changes in nuclear size in response to silibinin treatment, albeit with different sensitivities and different effects. S109 showed increased nuclear size (1.28-fold) at the highest dose (4.5 µM), while S114 and S199 showed a decrease in nuclear size (-1.6-fold and -1.9-fold respectively) at the same dose. Both S322 and S438, interestingly, showed significant changes in size only at lower doses (1.36-fold and 1.2-fold), suggesting a potential biphasic response which was unanticipated based on the dose-response curve for 786-O (Figure 3Q). Lastly, FDI-6 showed the most consistent response between lines tested, all showing a dose-dependent decrease in nuclear size (Figure 8I,J). Some lines were more sensitive to treatment than others; however, the dose predicted from 786-O induced a significant, non-toxic effect in all lines (less than 5% stained with Ethidium Homodimer). Interestingly, as observed with 786-O, the high-dose treatment with FDI-6 induced morphological changes in many cells, including loss of Hoechst 33342 staining (Figure 8K). These colonies with degenerated nuclei are excluded from the nuclear morphology analysis because it is based only on Hoechst 33342 stained cells, and our quantification methods may therefore underrepresent the inhibitory effect of FDI-6 treatment on tumor colony formation. Our analyses show that silibinin and FDI-6 target pathways that profoundly modify cell behavior and tumor colony growth in patient tumor replicas, identifying them and their related compounds as interesting candidates for further investigation in ccRCC models. Furthermore, dose titration using 786-O provided relevant starting doses for treatment of patient-specific tumors in 3D cultures.

## 4. Discussion

The screen developed in our study identified three candidate compounds that replicate aspects of the cell cycle phenotype caused by inactivation of *FOXD1* in 786-O cells. Using patient-derived tumor replicas, we demonstrate that the selected compounds perturb growth in multiple primary tumor lines. The 3D culture method that we developed to mimic the ECM environment of ccRCC provides a sensitive platform to study treatment responses and may provide important information on patient-specific tumor sensitivity to panels of compounds. At the doses tested, we found that subsets of patient replicas were sensitive to silibinin, while sensitivity to FDI-6 appears more general across patient tumor replicas. In addition to identifying these factors as compounds of interest for further studies in tumor models, our analysis provides a foundation for a screening method that could be used to select candidate treatments specific to a patient’s tumor. Further development of this concept will include the addition of tumor stroma elements, which our previous studies have shown are maintained in this culture system [9].

We have previously shown that loss of FOXD1 dysregulated phosphorylation of CDC2 (which is essential for progression through the G2/M checkpoint) and histone H3. Two of the three candidates identified in our study are known to have effects on CDC2 and Histone H3. FOXM1 has been shown to directly regulate CDC2, with its loss impairing CDC2 activity [51]. CDC2 is also regulated by the phosphatase PP2A [52], which is inhibited by PME-1. Additionally, PP2A also regulates phosphorylation of Histone H3, giving more credence to the role of PME-1 in the G2/M growth delay phenotype.

FOXM1 is a master regulator of the cell cycle and regulates the passage of cells through S and G2/M [53]. It has been implicated in several cancer subtypes, including RCC, and is highly prognostic [36]. Loss of FOXM1 has been shown to lead to mitotic decline, senescence, and necrosis in both aging and in cancer treatment with FDI-6 [54,55]. FDI-6 has been shown to cause nuclear fragmentation and subsequent dissolution in cancer cells [55]. We report this effect occurring in an immortal cell line, as well as in primary tumor cells lines. Revisiting the initial RNA-seq, GSEA analysis identified a cluster of pathways related to cell senescence, necrosis, and apoptosis that are upregulated with loss of *FOXD1* (Appendix A). Induction of senescence in tumors is an attractive therapeutic strategy, and the possibility that FDI-6 induces senescence in ccRCC cells will be explored in future studies.

PME-1 is a methylesterase that binds to the active site of protein phosphatase 2A (PP2A), resulting in demethylation and inactivation. It has been shown to influence formation and elongation of the mitotic spindle, with inhibition by AMZ-30 resulting in mitotic crises and cell death [56]. In this analysis, we see that inhibition of PME-1 did not completely inhibit the ability of cells to create colonies but did result in increased mitotic defects similar to the loss of *FOXD1* in 786-O cells. Long-term inhibition of PME-1 would likely result in gradual growth inhibition with the accumulation of mitotic defects. However, in primary lines tested in this study, AMZ-30 was observed to have little effect on tumor colony growth or nuclear morphology. Re-titration of AMZ-30 in each patient line and investigation of the effects of other inhibitory compounds will address if PME-1 is a useful target to modify ccRCC tumor growth.

Little is known about TMEM167A, but it has prognostic significance in lung cancer and glioma [57,58]. TMEM167A regulates vesicular trafficking, TNF and EGF signaling, and p53 transcription [29]. The compound silibinin has been shown to inhibit the TMEM family of receptors [57,58]. While being the least specific, it has known antitumor effects [59]. Further investigation of the targets of silibinin in ccRCC cells will be required to understand if its growth inhibitory effect is due specifically to the inhibition of TMEM167A.

Empirical determination of the specificities of compounds selected from our screen was beyond the scope of this study, in which the main aim was to find compounds that recapitulate the phenotypic effect of *FOXD1* inactivation. Further investigation of candidate compounds will be required to identify their precise mechanisms of action. If we confirm that they target independent pathways, it is possible that combined treatment of cells will result in synergistic effects that may more closely replicate the *FOXD1* loss of function phenotype with higher fidelity than single compounds.

Our analysis of FOXD1 targets revealed three therapeutically interesting candidates and demonstrated profound effects of their growth inhibition on both 786-O and primary patient tumor cells. Although our study did identify other candidates with effects on the cell cycle, we chose to focus on FOXM1, PME-1, and TMEM167A because they recapitulated the G2/M delay that we previously characterized in *FOXD1* loss of function cells. The profound change in growth characteristics of tumor cells caused by loss of *FOXD1* makes it an attractive therapeutic target, and since there is no reported strategy to inactivate FOXD1 itself, our work focused on the downstream pathways that are targetable. Considering the success of our approach in identifying potent targets, it could be expanded using large compound libraries in conjunction with our tumor replica modeling to develop a high throughput screen of candidate treatments for ccRCC. A challenge with the current study is the dependence on microscopy to observe changes in nuclear morphology, which limits the ability to test a large range for dose-response or to analyze many replicates. Utilization of an automated imaging system may overcome these limitations.

Given the tractability of outgrowth of primary patient material in our tumor replica cultures, an intriguing possibility for further development would be to compare compound responsiveness of primary versus metastatic ccRCC tumors from the same patient. While this is a technically challenging study due to the limited availability of patient material, it would provide clinically important insights into differences in tumor cell treatment sensitivity that arise following seeding to a non-renal location.

## 5. Conclusions

In this work we have developed an analysis strategy to discover pathways downstream of FOXD1 that control the G2/M phase of the cell cycle in ccRCC using a 786-O^FOXD1null^ ccRCC cell line. This strategy generated a list of actionable candidates that could be experimentally modulated. Eighteen candidate compounds were tested, of which three were able to copy aspects of the *FOXD1* loss of function phenotype: FDI-6 (against FOXM1), AMZ-30 (against PME-1), and silibinin (against TMEM167A).

A novel 3D culture system that replicates the ccRCC environment was used to test the capacity of selected compounds to inhibit tumor cell growth, and strong growth inhibitory effects were observed for silibinin and FDI-6. The 3D culture system enabled testing of patient-derived tumor cells, which showed varying susceptibility to the candidate compounds. In addition to the identification of target pathways for further investigation as ccRCC tumor growth inhibitors, the study identified significant variability in responsiveness to inhibitory compounds between patient-derived tumor models, suggesting that personalized tumor replicas may be an important addition to a discovery pipeline at an early stage.

## Figures and Tables

**Figure 1 cancers-14-03958-f001:**
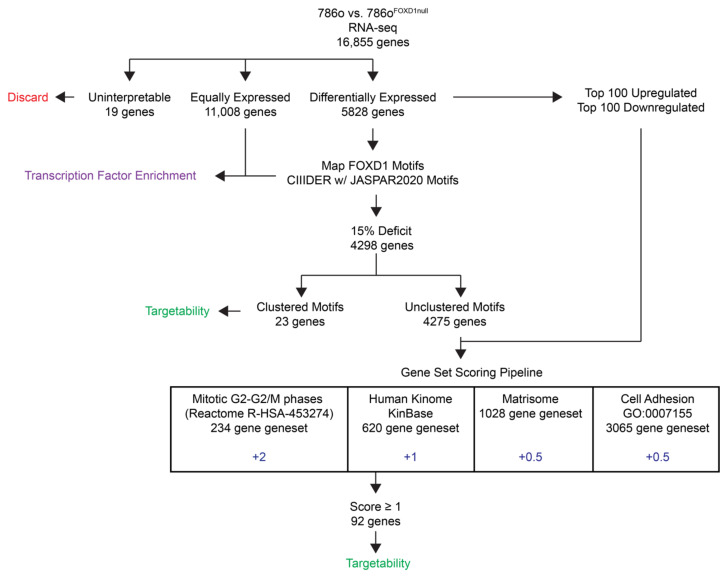
RNA-sequencing analysis and candidate selection.

**Figure 2 cancers-14-03958-f002:**
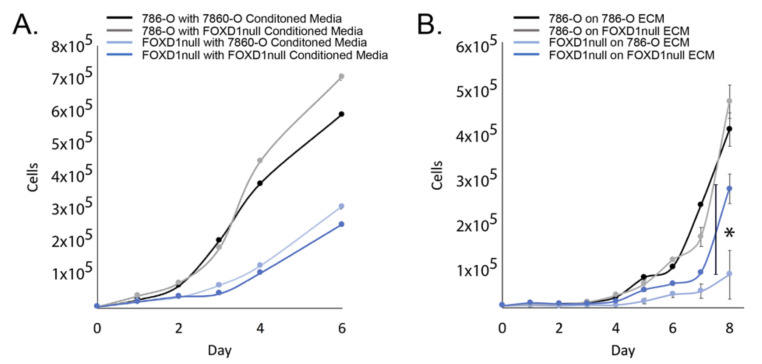
Exclusion of secreted factors and ECM from target discovery. (**A**) growth rate analysis of 786-O and 786-O^FOXD1null^ grown in conditioned media from 786-O or 786-O^FOXD1null^; (**B**) growth rate analysis of 786-O and 786-O^FOXD1null^ grown on native decellularized ECM from 786-O or 786-O^FOXD1null^. * = *p*-value < 0.05 by Student’s *t*-test at end-point.

**Figure 3 cancers-14-03958-f003:**
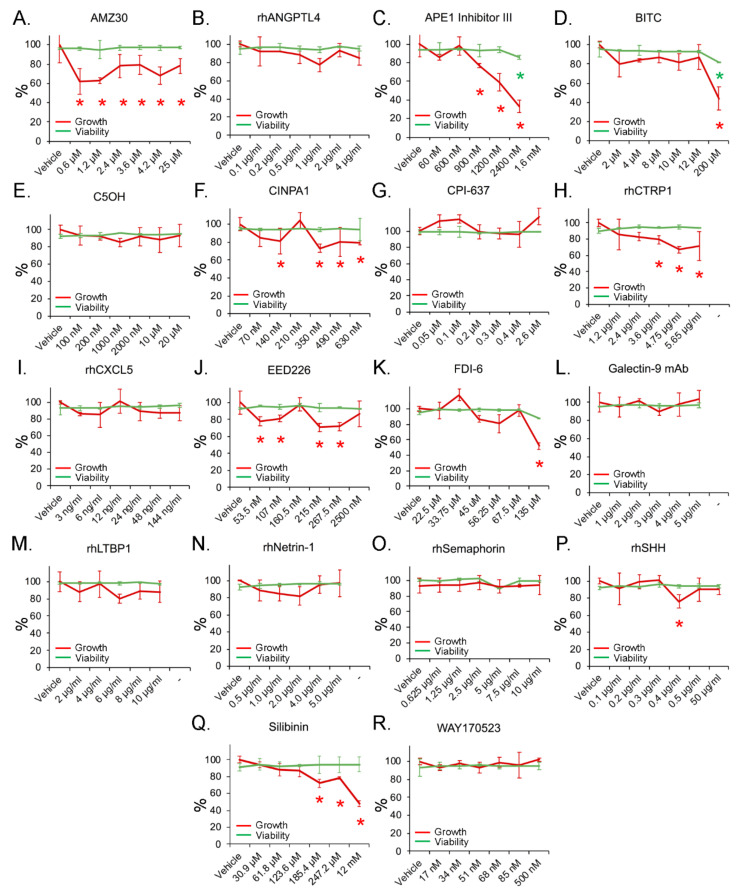
Cell growth and viability screen of compound-treated 786-O cells cultured for three days. (**A**–**R**) cells treated with compounds at doses specified on *x*-axis. Percentage growth (red line) was calculated based on the average number of Hoechst 33342^+^ cells in each condition. Percentage viability (green line) was calculated based on the average proportion of Ethidium Homodimer-1 positive nuclei to negative nuclei. Color matched asterisks indicate significance compared to vehicle control, as determined by a Student’s *t*-test.

**Figure 4 cancers-14-03958-f004:**
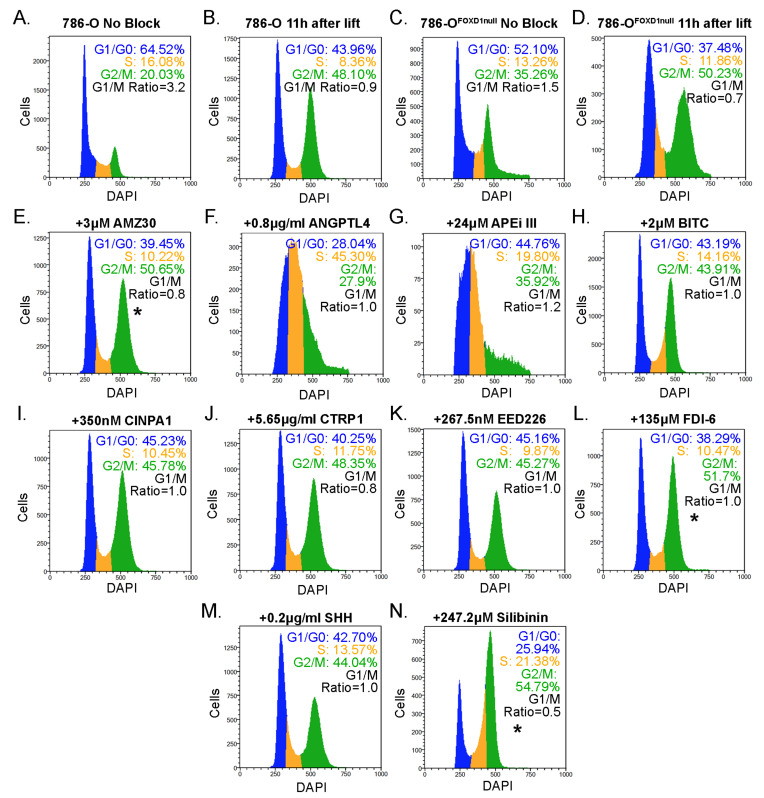
Synchronized cell cycle analysis identifies targets that inhibit G2/M progression. (**A**–**D**) untreated controls were compared to non-synchronized cells to confirm that the blocking and lifting strategy worked as predicted. (**E**–**N**) synchronized cells were treated with indicated treatment upon removal from the block and allowed to grow for 11 h. Fixed cells were analyzed for cell cycle distribution using DAPI, and the distribution of cells at G2/M was compared to untreated. * indicates *p* < 0.05 by a Student’s *t*-test.

**Figure 5 cancers-14-03958-f005:**
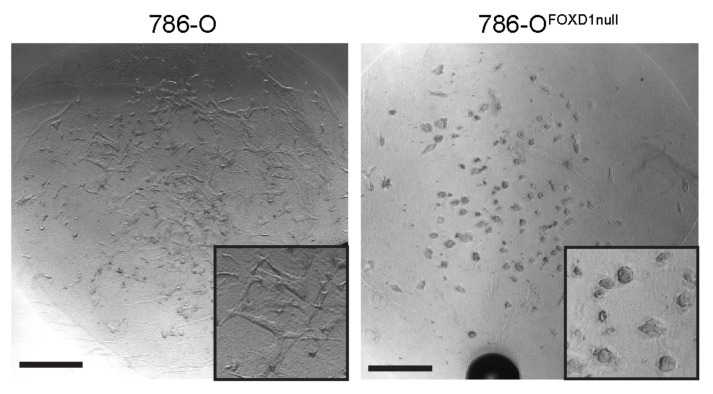
786-O 3D growth. 786-O (**left**) or 786-O^FOXD1null^ (**right**) 3D cultures grown in fibrin/Tumor ECM. Inset images show higher magnification to detail structural differences in cell colony growth. Scalebar = 500 microns.

**Figure 6 cancers-14-03958-f006:**
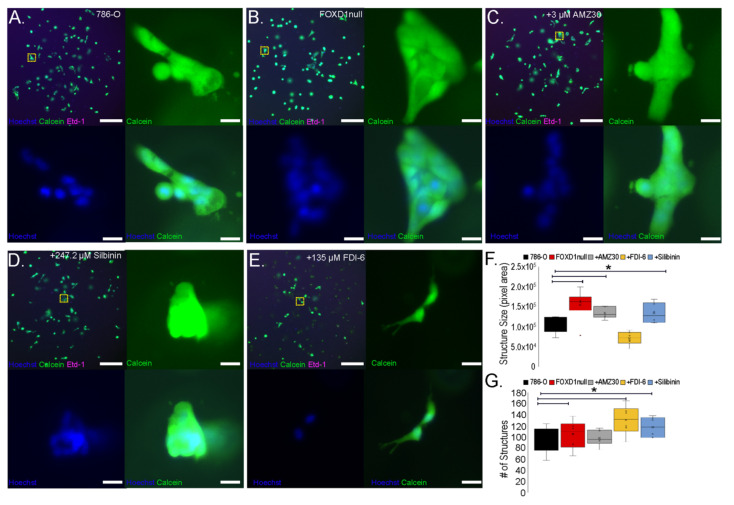
Effects of candidate compounds on 3D growth of 786-O tumor cells. (**A**–**E**) the 786-O 3D cultures were treated with specified compounds and allowed to grow for three days. Three-dimensional cultures were then stained for nuclei (Hoechst 33342 (blue)), viability (Calcein-AM (green)), and death (Ethidium homodimer-1 (magenta)). Upper left panels are widefield images (scalebar = 500 microns). Upper right, bottom left, and bottom right panels show split channels of representative cell colonies outlined in the widefield image (yellow box). The scalebar for higher magnification images is 20 microns. (**F**) the size distribution of Calcein-AM^+^ cell colonies was quantified. * = *p*-value < 0.05 calculated by Z-test. (**G**) The total number of Calcein-AM^+^ cell colonies in each culture was quantified. * = *p*-value < 0.05 calculated by Z-test. X = median, line through box = average.

**Figure 7 cancers-14-03958-f007:**
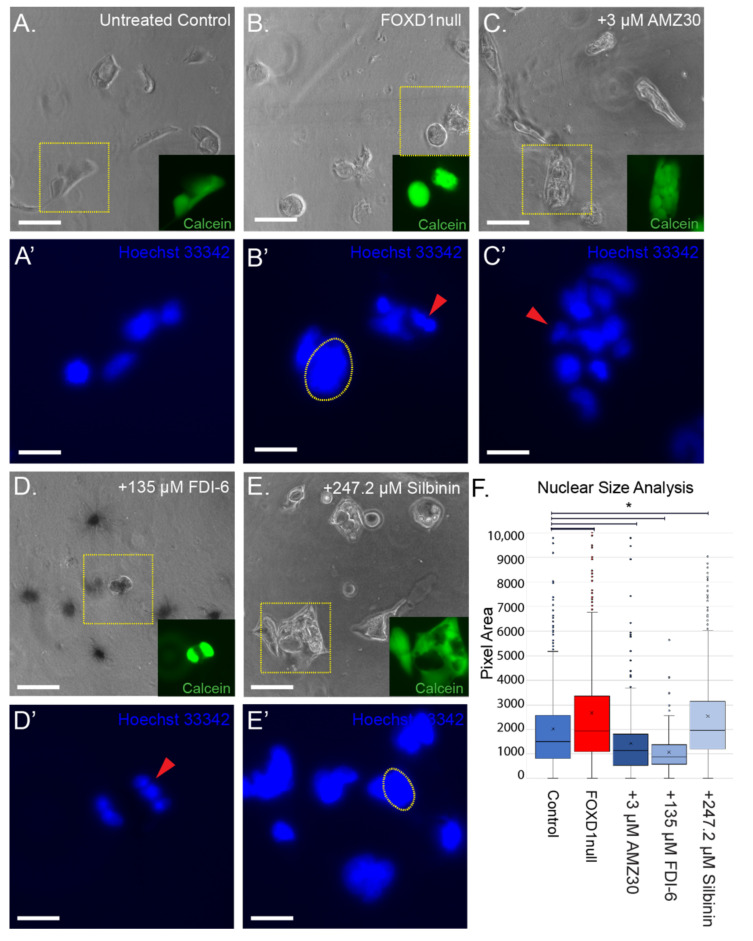
Effects of candidate compounds on nuclear structure in 786-O tumor cells grown in 3D culture. (**A**–**E**) images of 3D cultures treated with each candidate compound. Yellow box shows representative cell colony and inlay shows Calcein-AM (green) staining of that colony. Scalebar = 100 microns. (**A’**–**E’**) Hoechst 33,342 nuclear staining of cell colonies in yellow boxes in (**A**–**E**). Yellow-dotted outlines highlight enlarged nuclei. Red arrows indicate nuclear defects causing reduced nucleus size. Scalebar = 20 microns. (**F**) quantification of nuclear size in each treatment. * = *p* < 0.05 by Z-test. X = median, line through box = average. Circles outside the box are outliers.

**Figure 8 cancers-14-03958-f008:**
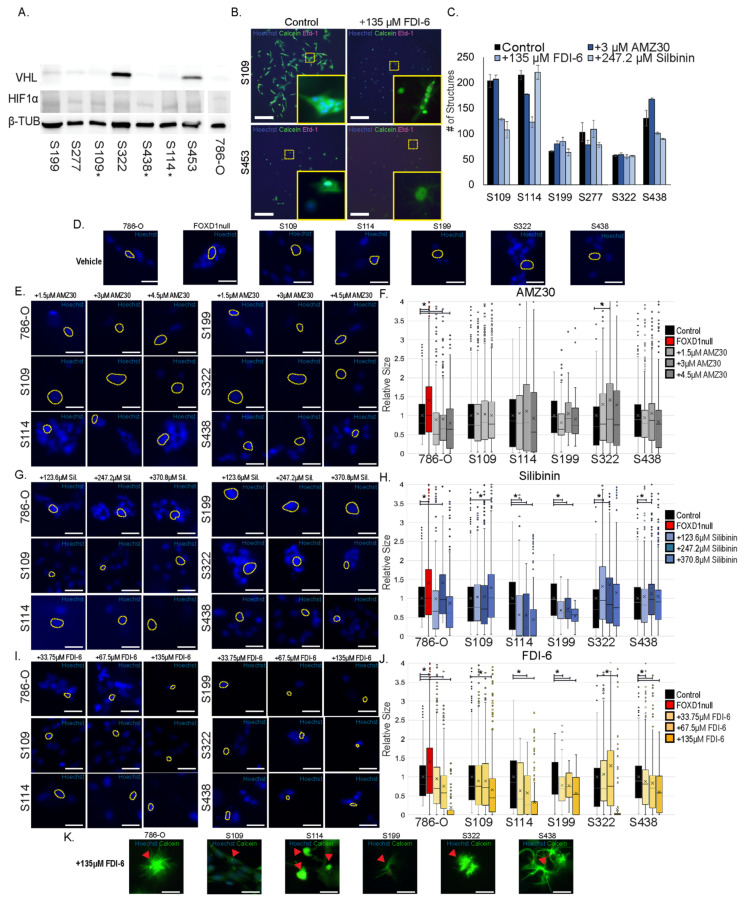
Test of candidate compounds on primary patient tumor replicas. (**A**) pseudo-hypoxia analysis of isolated primary cell lines and 786-O cells through Western Blotting (Appendix A uncropped Western Blot Figures). * indicates potential true VHL null. (**B**) representative images of 3D cultures after three days treatment with FDI-6. All images can be found in Appendix A. Three-dimensional cultures were stained with Hoechst 33342 (blue), Calcein-AM (green), and Ethidium homodimer-1 (magenta). Yellow-dotted box indicates region of inlay image. Scalebar = 500 microns (**C**) structure quantification of treated 3D cultures. (**D**) nuclei from colonies stained with Hoechst.33342 from each line tested. Yellow-dotted outlined nuclei are representative of each line (**E**,**F**). Nuclear morphology analysis of cells treated with indicated doses of AMZ30. Yellow-dotted outlined nuclei are representative of each treatment. Quantifications of nuclei sizes were normalized to average size from vehicle-treated control. * indicates significant difference in nuclear morphology compared to vehicle-control nuclei. (**G**,**H**) nuclear morphology analysis of cells treated with indicated doses of silibinin. Yellow-dotted outlined cells are representative of each treatment. Quantifications of nuclei sizes were normalized to average size from vehicle-treated control nuclei. * indicates significant difference in nuclear morphology compared to vehicle controls. (**I**,**J**) nuclear morphology analysis of cells treated with indicated doses of FDI-6. Yellow-dotted outlined cells are representative from each treatment. Quantifications of nuclei sizes were normalized to average size from vehicle-treated control nuclei. * indicates significant difference in nuclear morphology compared to vehicle controls. (**K**) representative cells from lines treated with a high dose of FDI-6. Red arrows indicate cells without Hoechst3342 staining.

**Table 1 cancers-14-03958-t001:** Tissue samples used in this study. NA = not applicable.

	Diagnosis	Stage	Grade	Race	Gender/Age
Tp17-s322	Unclassified renal neoplasm	NA	NA	White	F/70–79
Tp17-s438	Unclassified renal neoplasm	2	3	White	F/30–39
Tp18-S453	Clear cell renal cell carcinoma	1	3	White	M/80–86
Tp18-s109	Clear cell renal cell carcinoma	3	2	White	M/60–60
Tp18-s114	Clear cell renal cell carcinoma	3	3	White	M/80–89
Tp20-s199	Clear cell renal cell carcinoma	3	2	NA	M/74
Tp20-s277	Leiomyosarcoma	3	NA	White	M/69

**Table 2 cancers-14-03958-t002:** Candidate compounds identified from transcriptome analysis.

#	Compound	Target	Action	Reference
1	WAY 170523	MMP13	INHIBIT	[23,24,25]
2	EED226	SUZ12	INHIBIT	[26,27]
3	APE1 Inhibitor III	APEX1	INHIBIT	[28]
4	Silibinin	TMEM167A	INHIBIT	[29]
5	Recombinant Netrin	UNC5B	ACTIVATE	[30]
6	CINPA 1	CXADR	INHIBIT	[31]
7	BITC	AGPS	INHIBIT	[32,33]
8	AMZ-30	PME1	INHIBIT	[34]
9	FDI-6	FOXM1	INHIBIT	[35,36]
10	CPI-637	EP300	INHIBIT	[37]
11	Anti-Galectin-9 Antibody	LGALS9	INHIBIT	[38,39]
12	Recombinant Sonic Hedgehog/Shh	HHIP	INHIBIT	[40,41]
13	C5OH	S100P	INHIBIT	[25,42,43]
14	Recombinant C1qTNF1	C1QTNF1	ACTIVATE	[44]
15	Recombinant Angiopoietin-like 4	ANGPTL4	ACTIVATE	[45]
16	Recombinant CXCL5/ENA-7	CXCL5	ACTIVATE	[46]
17	Recombinant Semaphorin 3C	SEMA3C	ACTIVATE	[47]
18	Recombinant LTBP1	LTBP1	ACTIVATE	[48]

**Table 3 cancers-14-03958-t003:** Key FOXD1 related effects analyzed in 786-O cells. Summary of results from analysis of 3D growth of 786-O treated with chosen compounds compared to 786-O^FOXD1,^ with descriptions. + = less, ++ = more, +++ = most.

	786-O	FOXD1 null	+AMZ30	+Silibinin	+FDI6
# of Structures	**+**	**++**	**+**	**+++**	**++**
Colony Size	**++**	**+++**	**+++**	**++**	**+**
Colony Morphology	**Invasive**	**Round**	**Round**	**Round and Vacuolated**	**Irregular and protruding**
Nuclear Defects	**+** **Normal**	**++** **Large, mitotic defects prevalent**	**++** **Small and irregular**	**++** **Large and swollen**	**+++** **Very small, fragmented**

## Data Availability

RNA sequencing used in this study is available on GEO repository and accessible via the web (GSE210145).

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
