# Peer review of "Targets for Renal Carcinoma Growth Control Identified by Screening FOXD1 Cell Proliferation Pathways"

_cancers, 2022, doi:10.3390/cancers14163958_

Round 1
Reviewer 1 Report
This work presents a very novel strategy to identify tractable targets that are related to FOXD1 and important transcription factor in ccRCC.
I made the comments in the same pdf, The major concern is the dose response analysis. I think it can be improved.

Author Response
Responses to reviewers comments
Reviewer 1
We would like to thank the reviewer for their careful appraisal of our manuscript, and for their constructive comments. Below is an itemized list of responses; sections of the manuscript that have been revised are in red.
Text edits proposed at the following locations have been made:
P1L10
P1L11
P1L24
P2L47
P2L59
P2L62
P2L71
P4L47
P4L150
P4L152
P4L162
P4L164
P5L205
P7L280
P11L340
P13L391
P13L395
P15L463
P15L464
P15L484
P15L486
Comment: Regarding compound toxicity screen: Have you determined the IC50 or the ED50 in the cells you use in this work? Its not clear if the reported concentration in the Table 2 are reported by you or others. If not, this could be more clear: 1. Reported IC50… 2. Reported ED50… or 3. Reported working concentrations. For Table S2 I suggest adding the references and the type of cells used to determine the concentration.
Response:
IC50, ED50, or working concentrations were obtained through product datasheets or from references detailed in datasheets. To clarify this, Table S2 has been modified to include information regarding the source of the values, as well as the experimental validation context of those values. References have been added to Table 2 to cite sources in the main text (P9L337).
Comment: Regarding compound toxicity screen: Table 3 and Section 3.5: This section Is the part that I would suggest revising and improve clarity
- In the results text please add which compounds were actually significant inhibiting growth, adding the numbers.
- The viability/growth inhibition results should be better displayed as viability curves that can facilitate visually to see the effect of the compounds.
Response:
Table 3 has been replaced by a new Figure 3 (P10L347), which includes viability and growth curves for each tested compound, including the concentration range. The numbers listed in the table have been moved into the body of the results section text. This information, previously in Table S2 has been removed from that table. All other figure numbers have been adjusted accordingly.
Comment: Regarding compound toxicity screen: 1. How do you evaluate toxicity? 2. If the number of hoechst+ cells is reduced, does it mean that the cells actually died, disappear from the plates? They may have been Eth+ earlier?
Response:
There was a range in viability between vehicle-treated controls, with a maximum of 98.1% viability, an average of 94.14%, and a minimum of 88.7%. We determined the maximum tolerable dose to be a viability less than the minimum of 88.7% to account for this variability in viability, which was determined independent of the vehicle. Only two tested compounds, APE1 inhibitor III and BITC, showed toxic effects meeting this criterion. This indicates that for many of the compounds tested, there was strong tolerance by the 786-O cells. Considering that the maximum doses tested were at concentrations that may be considered unreasonably high (hundreds of times greater than the reported IC50 or EC50 values), we did not consider increasing the range further.
Viability was determined by the staining of Ethidium Homodimer-1, which is incorporated into nuclei of dying cells with compromised membranes, alongside Hoechst which is incorporated into all nuclei. To account for the possibility of dead or dying cells detaching from the plate, media was never removed for the entire time course. Viability staining was done by adding a 4X working concentration of dye solution to wells and imaged without any washing. Although we did not observe floaters in our images, they would have been captured and calculated alongside the attached cells.
Comment: How do you define the multicellular structures? What’s the minimum number of cells you consider to call them a structure different than a cell cluster that didn’t develop as a structure? Is there a cut-off?
Response:
Cells were seeded into structures as single cells, Multicellular structures were determined using the following criteria, which we have added to the methods section for clarity (P5L215): 1. More than 2 nuclei and 2. Are viable (Ethidium Homodimer-1-). Higher magnification images in Figure 6 show these structures in question (P6L44).
Comment: Regarding Figure 5: 1. You are missing the images of 786FOXD1null that you compared with in 5E and 5F. 2. Mark examples of structures that were measured using arrows.
Response:
The figure has been modified to include the images of the 786-OFOXD1null. Outlines and inlays have been added to each image as to show examples of viable structures measured in the analyses (P15L469).
Comment: Regarding treatment of patient-derived 3D cultures: Line 392: At what concentrations? Here I am not sure if the same dose range would apply than in the 2D commercial cell lines.
Response:
In order to answer this question, we repeated the 3D treatment experiments done on 786-O and primary cells with a 3-point titration of the chosen compounds (AMZ30, Silibinin, and FDI-6). We observed that there was a dose-dependent response to these compounds. We added these findings to the paper as part of Figure 8 (P18L557).
Comment: Regarding evaluation of patient-derived 3D cultures: 1. Also add numbers or fold changes when describing the results by how much the structures increase in number or the shape changed. 2. What type of morphological changes are described on P13L398? 3. Could the differences between response of patient-derived 3D cultures and 786-O be due to the drug concentrations being off? 4. In the discussion the statement regarding differential responses of patient lines is conditional to the dose range used.
Response:
All these comments have addressed as follows: fold changes have been added in the text (P13L415), the morphological changes have been described in the text (P13L424), and the response to the compound treatment has been addressed through titration of the compounds, as previously mentioned.
Comment: Materials and Methods section P2: Suggested re-writing for clarity: Microkit. RNA purity was determined by measuring the 260/280 ratio in a ThermoFisher Nanodrop. RNA quality was determined by running an RNAgel and analyzing the ratio between 28S and 18S.
Response: The text has been reorganized as recommended (P2L82).
Comment: Regarding conditioned medium experiment: How was the procedure to collect conditioned media. Add to methods. How many days at what confluency etc.
Response:
We added into the methods section details regarding the conditioned media experiment. We additionally added methods for the native ECM growth rate experiments as well. (P3L116)
Comment: Section: 3.6: I would add here in the text or in figure 3 the dose used for each drug.
Response: Text and figures have been modified accordingly.
Comment: P9L307 – clarify what comparator was use to determine that the three compounds showed an increase in distribution of cells at G2/M.
Response: Text has been modified to clarify this. (P11L363)
Comment: P11L328 Line 328: what do you mean by flat? it seems 786-O is flatter as they more elongated (invasive) while 786-O FOXD1null is round.
Response: As reviewer stated, it is more accurate to call the structures “round and expansive” and the text has been modified accordingly. (P11L378)
Comment: Figure S1 is missing concentrations and high mag inlays as well.
Response: Concentrations and inlays have been added.
Comment: Regarding Figure 7: 1. Add concentrations being used. 2. How do you quantify structures? Perhaps add it to the method section. 3. Also please add the methodology of quantifying nuclear size.
Response: Concentrations have been added to the figure. Additionally, methodology for quantifications was added to the methods section. (P5L215)
Comment: Add a summary diagram or table with the characteristics of FOXD1null and how the drugs compare in each category of 3D growth and morphology to describe how shape and number of structures relates to nuclear morphology?
Response:
A summary table was added (Table 3) to summarize the FOXD1null phenotypes and how chosen compounds effect it. (P16L486)
Comment: As it is described, it seems a bit dispersed the infromation on each drug. Some are similar in some aspects and some are simialr in others? Do some aspects/ parameters have more weight in terms of comparison to FOXd1null phenotype? or the 3 drugs are equally equivalent? In summary, section 3.8 needs a bit more grouping and interpretation of the results.
Response: The two main parameters we analyzed in this study were the total number of structures formed, and the nuclear morphology. The number of structures is an important parameter for 3D culture growth success, as we have previously reported [8]. The nuclear morphology is an important characteristic of FOXD1null cells, which show both increased distribution of cells at G2/M, as well as an increase in cell fragmentation [7]. Both of these events (increased nuclear size due to stalling at G2 and decreased nuclear size due to fragmentation) are important but potentially independent characteristics of the FOXD1null phenotype. Thus, we did not add weight to increased or decreased nuclear size. Additional text has been added to section 3.8 to contextualize this. (P11L384).

Reviewer 2 Report
Bond et al. present manuscript about targeting FOXD1 downstream targets by previously published compounds in ccRCC cells. FOXD1 was previously shown as important regulator of proliferation in ccRCC. Therefore it is interesting target for ccRCC therapy. Authors developed screening pipeline to identify FOXD1 affected genes and putative compounds that influence target genes expression. They also developed cell culture methods to test selected compounds. Generally paper is really interesting and well written. It also shows how creativity and literature screening could be used to find putative therapeutic targets and compounds that could be potentially used in treatment. I have only few concerns.
Major:
1. In the experiments, where Authors treated cells with selected compounds they used untreated cells as controls. I don't know how compounds were prepared, what kind of solvents were used, but if for example compound A is dissolved in DMSO, than control should be also treated with DMSO.
2. I cannot find description how 786-O FOXD1null was prepared. Please add description or cite the proper paper.
Minor:
1. Line 57 - probably Authors mean transcriptomic or transcriptome-wide analysis, not transcriptional analysis.
2. RNA gels should be attached to supplementary figures.
3. Lowercase 's' should be replaced by uppercase 'S' in 18S and 28S.
4. Full parameters list for Trimmomatic, STAR and featureCounts should be added.
5. Which software were used to perform GSEA? GSEA results should be further mentioned and discussed in Results section.
6. Line 94 – do you mean uniquely mapped reads?
7. Line 96 - it looks like filtering was done based on adjusted p value, description should be corrected.
8. Some information about renal carcinoma (occurrence, treatment strategies) should be added to Introduction.
9. Limitation of the study should be clearly pointed out in the Discussion.
10. It should be pointed out in Discussion that determining compound(s) specificity is beyond the scope of the paper and the main aim was to repeat phenotypic effect caused by FOXD1 KD using compound(s) treatment.
Author Response
Responses to reviewers comments
Reviewer 2
We would like to thank the reviewer for their careful appraisal of our manuscript, and for their constructive comments. Below is an itemized list of responses; sections of the manuscript that have been revised are in red.
Comment: In the experiments, where Authors treated cells with selected compounds they used untreated cells as controls. I don't know how compounds were prepared, what kind of solvents were used, but if for example compound A is dissolved in DMSO, than control should be also treated with DMSO.
Response:
For each compound tested, a vehicle control was included at comparison of the highest dose. For example, for the compound Silibinin, which was reconstituted in Methanol, the highest dose tested was 12mM. The vehicle control for this included an identical amount of methanol matching this highest tested dose. No significant difference between vehicle controls was calculated.
Comment: I cannot find description how 786-O FOXD1null was prepared. Please add description or cite the proper paper.
Response: 786-OFOXD1null was generated by mutating the FOXD1 gene using a targeted crispr/cas9 approach which is described in Bond KH, Fetting JL, Lary CW, Emery IF, Oxburgh L. FOXD1 regulates cell division in clear cell renal cell carcinoma. BMC Cancer. 2021 Mar 24;21(1):312. This information was inadvertently omitted and has been included in the Marerials and Methods section on P2L71.
Comment: Line 57 - probably Authors mean transcriptomic or transcriptome-wide analysis, not transcriptional analysis.
Response:
We meant transcriptomic. The text has been modified (P2L60).
Comment: RNA gels should be attached to supplementary figures.
Response:
Gels have been included in the gel submission.
Comment: Lowercase 's' should be replaced by uppercase 'S' in 18S and 28S.
Response:
Text has been modified (P3L82).
Comment: Full parameters list for Trimmomatic, STAR and featureCounts should be added.
More information regarding these parameters, as supplied by Genewiz, was added to the methods section (P3L88).
Response:
Text edits have been made accordingly (P3L88).
Comment: Which software were used to perform GSEA? GSEA results should be further mentioned and discussed in Results section.
Response:
We used the GSEA Analysis software from The Broad Institute. Citation has been added (P3L98). The GSEA analysis was included in Figure S2 and mentioned in the discussion section.
Comment: Line 94 – do you mean uniquely mapped reads?
Response:
Yes, the text has been modified accordingly (P3L102).
Comment: Line 96 - it looks like filtering was done based on adjusted p value, description should be corrected.
Response:
Yes, text has been corrected (P3L104).
Comment: Some information about renal carcinoma (occurrence, treatment strategies) should be added to Introduction.
Response:
A brief discussion has been added to the introduction (P2L44).
Comment: Limitation of the study should be clearly pointed out in the Discussion.
Response:
Some limitations of this study was added into the discussion section (P19L586).
Comment: It should be pointed out in Discussion that determining compound(s) specificity is beyond the scope of the paper and the main aim was to repeat phenotypic effect caused by FOXD1 KD using compound(s) treatment.
Response:
Text has been added to the discussion to address this (P19L591).

Round 2
Reviewer 2 Report
Generally Authors addressed all my comments.
I only found some minor typos, that should be corrected:
Line 47 – citation should be added
Line 84 - is ‘smples (28s/18S > 2.0)’, should be 'samples (28S/18S)'
I also cannot find supplementary figures cited in the text. Only table s2 is present, when I download supplementary data.
Lastly, GEO accession number is in my opinion absolutely required in the final version of the manuscript.
